# Tree Species and Morphology of Holes Caused by Black-Tufted Marmosets to Obtain Exudates: Some Implications for the Exudativory

**DOI:** 10.3390/ani12192578

**Published:** 2022-09-27

**Authors:** Juliane Martins Lamoglia, Vanner Boere, Edgard Augusto de Toledo Picoli, Juraci Alves de Oliveira, Carlos de Melo e Silva Neto, Ita de Oliveira Silva

**Affiliations:** 1Pos Graduate Program at Animal Biology, Universidade Federal de Viçosa, Viçosa 36570-900, MG, Brazil; 2Instituto de Humanidades, Artes e Ciências, Campus Jorge Amado, Universidade Federal do Sul da Bahia, Itabuna 45653-970, BA, Brazil; 3Post Graduate Program at Environmental Sciences and Technology, Universidade Federal do Sul da Bahia, Porto Seguro 41900-222, BA, Brazil; 4Departamento de Biologia Vegetal, Universidade Federal de Viçosa, Viçosa 36570-900, MG, Brazil; 5Department of General Biology, Universidade Federal de Viçosa, Viçosa 36570-900, MG, Brazil; 6Instituto Federal de Goiás, Cidade de Goiás 76600000, GO, Brazil

**Keywords:** *Callithrix penicillata*, Cerrado, dendrometry, *Croton urucarana*, feeding ecology, *Tapirira guianensis*

## Abstract

**Simple Summary:**

Black-tufted marmosets (*Callithrix penicillata*) regularly exploit exudates by gouging the bark from trees with their specialized teeth. Determining preferred tree species to exploit exudates is important for forest management aimed at maintaining urban populations of marmosets. We investigated the trees used by black-tufted marmosets to obtain exudates in urban forest fragments in the Brazilian Cerrado, with a view to better understanding the relationship between marmosets and obtaining exudates. We characterized the tree species, the dendrometry, and the characteristics of the holes made by the marmosets. Based on these data, we proposed a gouging effort index (ChiSI). We found 108 individuals that were exploited by black-tufted marmosets belonged to 16 species of 10 tree families. Eleven new tree species used by black-tufted marmosets for exudates were identified. Exudate exploration was predominantly of medium intensity, with round holes concentrated in the canopy. The species *Tapirira guianensis* and *Croton urucarana* were preferred. Therefore, *T. guianensis* and *C. urucarana* should be considered the most important tree species for black-tufted marmosets that live in urban forests in the Cerrado.

**Abstract:**

Knowledge of the pattern of exploitation of trees can help us understand the relationship between marmosets and plants, especially in declining forests, such as those in the Brazilian Cerrado. Black-tufted marmosets (*Callithrix penicillata*) regularly exploit exudates by gouging the bark of trees with their specialized teeth. Determining preferred tree species to exploit exudates is important for forest management aimed at maintaining an essential food source for urban marmosets. We characterized the tree species, dendrometry, and the characteristics of the holes made by marmosets to obtain exudates. Based on these data, we proposed a gouging effort index (Chiseling Suitability Index for Marmosets, ChiSI). We identified 16 species belonging to 10 families of trees with gouging marks made by marmosets. Eleven new tree species used by black-tufted marmosets for exudates were identified in urban forests in the Cerrado. Exudate exploration was predominantly of medium intensity, with round holes concentrated in the canopy. The species *Tapirira guianensis* and *Croton urucarana* were preferred. The ChiSI was characterized by a narrow range for both *T. guianensis* and *C. urucarana*. Despite the flexibility of obtaining exudates, the black-tufted marmosets concentrate their exploitation to only a few tree species. The *T. guianensis* and *C. urucarana* tree species should be considered the most important species for management plans and the preservation of black-tufted marmosets that live in urban forests in the Cerrado.

## 1. Introduction

Black-tufted marmosets (*Callithrix penicillata*) are primates of the Callithrichidae family that have spread throughout urban environments, occupying small, forested areas and orchards [1]. Their small size and a diet based almost exclusively on exudates allow the marmosets to live in small areas, making them less vulnerable to the effects of fragmentation [2]. Marmosets can therefore colonize and reproduce successfully in secondary forest environments and urban fragments [3,4].

The marmoset’s diet is based mainly on tree exudates, fruits, and animal prey [5]. Exudativory is one of the most important ecological adaptations of the genus *Callithrix*, although there are different levels of dependence on exudates for food between marmoset species. While fruits and animal prey can be unpredictable, exudates are a regular food source, exploited through the specialized dentition of marmosets (well-illustrated in the book of Hershkovitz, 1977), particularly in marmosets of the species *C. penicillata* and *Callithrix jacchus* (common marmoset), who are able to make holes in the bark of trees [6].

Marmosets of the *C. jaccchus* species have a bite force of up to eight times their body mass and have a wide jaw opening. The lower incisor teeth of marmosets are in a plane in line with the canine teeth, facilitating gnawing with the jaw. In addition to these characteristics, the lower teeth of marmosets are more resistant to the stress of gouging, due to their histological characteristics, such as the lower incisor enamel being thicker labiolingually, larger roots in relation to the symphyseal bone volume, and the surface area of the greater root in relation to its symphyseal volume (Vinyard et al., 2009).

The black-tufted marmoset, which lives in the Cerrado biome, actively exploits exudates from a variety of tree species commonly found in many habitats, such as riparian forests, Cerradão forests, and urban forest fragments [7,8,9,10]. Several factors can interfere with the exploitation of trees by marmosets to obtain exudate. Some studies have related this to a preference of some species to gouge trees that are more abundant or that have morphological and physiological characteristics that allow for efficient exploration of exudates [7,11,12]. Morphological aspects can be evaluated by dendrometry, such as tree height, diameter at breast height (DBH), circumference at breast height, basic density (the ratio between the absolute dry weight of the wood, and its volume, while in a state of complete water saturation. g·cm^−3^), and bark hardness [7,12,13,14]. Patterns of exploitation of exudate trees, such as the form and location of holes, can elucidate some aspects of a marmoset’s preference for tree species [12,15,16]. For example, holes caused by marmosets should be of a size and shape that can shed a sufficient volume of exudate. Therefore, the oral apparatus of the marmosets must be capable of gouging the bark of trees [6].

It is more difficult to estimate the cost of obtaining exudates, and therefore, this effort and energy cost are neglected in most studies. Some authors have suggested that the position of the body during the exploitation of exudates increases vulnerability to predators, influencing the duration and location of exudate feeding [17]. However, we did not find other studies that estimate the effort required by marmosets to obtain exudates. It is likely that direct data on the effort required by free-ranging marmosets to obtain exudates has operational limitations that are difficult to solve. While an assessment of the metabolic cost is impractical, part of the marmoset’s effort is linked to intrinsic properties of the bark, such as its density and thickness. The marmoset must exert effort to overcome the physical barrier of the tree’s bark. Therefore, indirect estimates, such as the effort to remove parts of the bark to reach the exudate ducts of the tree, may be used.

The region of the Cerrado is under intense anthropogenic pressure, rapidly losing its native vegetation cover and putting its fauna at risk of disappearance [18]. The ongoing and expected global climate changes over the coming decades will contribute to deep ecological changes in the Cerrado, though little is known about the relationship between primates and the flora in this ecosystem. The study of plant species found in different environments, as well as analysis of the interaction between animals and trees, allow for a better understanding of the ecology of exudate-exploiting primates. Therefore, we described aspects of exudativory such as tree-species preference, location, and the holes’ morphological traits, such as lesions caused by black-tufted marmosets in forest fragments of a large city. In other words, we described aspects of marmoset exudatives in forest fragments of a large city. Based on our findings, we also propose a “gouging effort index”, which helped us to better understand the choice of trees for exudativory by marmosets.

## 2. Materials and Methods

### 2.1. Study Areas

We carried out this study in urban forest fragments of the Cerrado, in the municipality of Goiânia, Goiás, Brazil. In the city, there are still small, forested areas with vegetation characteristic of the Cerrado habitat, although some exotic tree species have been introduced in the past. The predominant climate in the region is tropical, with a dry season of type Aw (dry winters) according to the Köppen’s classification, and it is characterized by two very distinct seasons: a cold and dry winter and a hot and rainy summer, with an average annual precipitation of 1520 mm, and an average annual temperature of 23.1°C [19].

Initially, we characterized the phytophysiognomy and the presence/absence of marmosets in 14 forest fragments. Three areas with marmoset groups were randomly selected for the study: Taquaral Municipal Park (S16°41′54″; W49°20′46″), the forest of the Madrid Gardens Residential Condominium (S16°45′3″; W49°20′43″), and the forest of the Memorial Museum of the Cerrado (S16°44′15″; W49°12′52″). All of these fragments were surrounded by houses, had free access for people, and were characterized as riparian semi-deciduous forests associated with streams.

### 2.2. Data Collection and Analysis

Data were collected from June 2014 to September 2015. All trees containing holes made by marmosets were marked with sequentially numbered aluminum plates and subsequently identified. The trees were identified by a specialist in the field, using specific bibliography [20]. The position of each sampled tree was georeferenced with the aid of a GPS device (Garmin, model GPSMAP 76CSx). To verify the number and species of trees gouged by marmosets, estimates were made on the relative frequency (in percentage) per fragment.

We measured the DBH and tree height with a clinometer (Haglöf Electronic). Trees were classified as gouged when at least one hole made by a marmoset was found with recent hole-gouged characteristics (i. e., with wet and liquid exudate). Trees without holes or with completely closed scars were characterized as intact (recovered). The intensity of exploitation by marmosets was divided into three classes based on the number of holes present in each tree: light (≤10 holes), moderate (11–50 holes), and intense (>50 holes) [11]. We recorded the locations of the holes in the trees (trunk or canopy). We also characterized the holes by shape: Round (similar vertical and horizontal diameters); Elongated (larger in one of the dimensions, either horizontal or vertical); or Irregular (no defined shape) (Figure 1).

We collected bark samples from the two most commonly used plant species in each forest fragment at breast height, with a depth reaching the cambium, for further analysis of density and thickness. The density of the bark was obtained from the ratio between the dry mass and the saturated volume, and the final thickness was measured with the aid of a digital caliper (6” Diamond Vernier Digital, Mitutoyo, Japan). The wood density classification followed the MB 26/40 standard of the Brazilian Association of Technical Standards [21].

To verify the suitability of the tree bark for the marmoset’s ability and capacity to scarify, the Chiseling Suitability Index for Marmosets (ChiSI) was proposed, which is the product of thickness (T) and density (De) of the bark of the trees. The formula for ChiSI is:ChiSI = T × De

This formula considers the thickness (distance from the outside of the shell to the cambium) and the density (ratio of dry mass/green volume in g/cm3) [22] (Avery & Burkhart 2001). The ChiSI makes it possible to determine the preferences for scarified trees, according to the capacity and effort required to produce a hole with exudates. We applied a descriptive statistical analysis to show the data using the PAST (Paleontological Statistics) free software, version 4.03 [23]. Regulatory committee authorization was not required since these studies were not conducted in public conservation areas or on plant species protected by law. Permission was granted by the managers of the private areas where the samples were collected.

## 3. Results

In the 14 forest fragments visited, we are able to identify 16 species belonging to 10 families of trees used by black-tufted marmosets. In three urban fragments chosen randomly for analysis and sore pattern, we identified 11 new tree species as sources of exudates (Table 1). Two species had inconclusive identifications due to the difficulty in analyzing the phenotypic characteristics. In all, 108 tree individuals were sampled (Table 1), among which, 76% belonged to the species *Tapirira guianensis* Aubl. (Anacardiaceae) (Figure 1) and *Croton urucurana* Baill. (Euphorbiaceae) (Figure 2). The Fabaceae family had the greatest variety of species that were used by marmosets (Table 1).

The intensity and location of exploitation on the tree varied among the species. In 53% of the trees, we recorded holes exclusively in the canopy, in 23.5%, on the trunk, and in 23.5%, on both the canopy and the trunk. Regarding the shape of the holes, 76.5% were round and 23.5% were combined shapes (elongated or irregular) on the same tree. Most tree species were moderately exploited (52.9%), followed by an equal percentage (23.5%) of high or light exploitation.

The marmosets in this study scarified trees with a bark density between 0.34 ± 0.06 g/cm3 (*Anacardium occidentale*) and 0.76 ± 0.08 g/cm3 (*Anadenanthera* sp.) (Table 2), having a bark thickness ranging from 3.11 ± 0.80 mm (*Mimosa caesalpiniaefolia*) to 11.51 ± 3.74 mm (*Anadenanthera* sp.). Despite these variations, the marmosets preferred to concentrate their scarifications on trees with a mean density between 0.50 ± 0.02 g/cm3 (*C. urucurana*) and 0.66 ± 0.04 g/cm3 (*T. guianensis*). These trees had an average bark thickness of between 4.89 ± 0.02 (*C. urucurana*) and 5.07 ± 0.54 mm (*T. guianensis*). The marmosets concentrated their exploitation of trees between those having a ChiSI value of 2.35 (*T. guianensis*) and 2.44 (*C. urucurana*) (Table 2).

Considering that the marmosets mostly used the two species *T. guianensis* and *C. urucurana*, our analysis focused on these. *T. guianensis* trees had a mean (±standard deviation) DBH of 25.25 cm ± 7.50 and a mean height of 14.2 m ± 4.65. *C. urucurana* individuals had a mean DBH of 16.98 cm ± 7.12 and a mean height of 11.1 m ± 2.97. The main shape of the holes was round and most of them were concentrated in the canopy of both tree species, *T. guianensis*, and *C. urucurana*.

## 4. Discussion

In addition to seven species already described [7,24,25], we identified eleven new tree species exploited by black-tufted marmosets to obtain exudates. Among the diversity of trees recorded, the families most often used were the Anacardiaceae, Fabaceae, and Euphorbiaceae. Similar studies on black-tufted marmosets also found a higher frequency of exploitation of the Anacardiaceae and Fabaceae families [7,24,25], but not of the Euphorbiaceae, which is a family recognized as the one most used by primates as a source of exudates [25].

According to some authors [7,9,10], tree species are used to obtain exudates in proportion to their total abundance within an area. In contrast to studies carried out in the Central Plateau (Planalto Central) [7,8,9,11], the Vochysiaceae family was found to be little exploited by black-tufted marmosets in this study. In that region, trees of the Vochysiaceae family had a greater relative abundance [7,8,9,11], while in this study the Anacardiaceae and Euphorbiaceae families were more abundant (Table 1). Our results are in agreement with other studies [7,8,9] that demonstrated that black-tufted marmosets exploit more abundant exudate trees in an area. According to other studies, the variety of trees used to obtain exudate demonstrates the food-exploitation flexibility of black-tufted marmosets [7,8,9,10]. The use of tree species that are concentrated in an area makes it possible to increase the availability of a food source with the least energy expenditure required for searching over long distances, constituting an adaptive mechanism for food ecology in fragmented environments [5].

*T. guianensis* and *C. urucurana* are fast-growing native pioneer tree species, frequently found in secondary formations and adapted to soils that temporarily flood. Both *T. guianensis* and *C. urucurana* are species recognized to be efficient in the regeneration of degraded areas [26]. *T. guianensis*, widely known as pau-pombo, is a native tree found in almost all forest formations and widely distributed throughout the Brazilian territory [26].

*T. guianensis* was the most relevant tree species for exudate exploitation by black-tufted marmosets, concurring with what has been described in other studies [7,8,9,10]. Other studies described the active exploitation of *T. guianensis* as a source of exudates by *C. jacchus* [5]. It was observed that *C. geoffroyi* and *C. kuhlii* only exploit *T. guianensis* exudates opportunistically because these marmosets have a weak capacity to gnaw on the bark [4,27]. The second most-utilized species was *C. urucurana*, popularly known as sangra-d’agua, which was reported for the first time as an exudate source for black-tufted marmosets. *C. urucurana* is widely distributed and occurs predominantly in riparian forests of various forest formations or on floodplains in humid, swampy soils that are subject to periodic flooding [26].

The way in which tree species were used varied interspecifically, possibly due to the anatomical characteristics of the bark of each species. Species with harder and thicker bark were apparently avoided by marmosets, as suggested in another study [7]. Tree species that had either a thorny trunk, rhytidoma that flake off, or very thick bark, were only scarified in the canopy, as verified with the species, *Anadenanthera* sp., *Piptadenia gonoacantha*, *Triplaris gardneriana*, and *Zanthoxylum rhoifolium*. Preference for the canopy regions was observed with holes made by hybrid marmosets in the *Anadenanthera peregrina* species from the Atlantic Forest, which has a very thorny trunk [12]. In this case, the canopy has a thinner bark and no thorns, facilitating scarification and access to the tissues where exudates are formed. Furthermore, due to a greater photosynthesis, the tree’s metabolism is more intense in the canopy, suggesting the probability of greater exudate production.

Most of the holes were round, with elongated or irregular holes found less frequently. The predominance of round-shaped holes did not appear to be random. Variation in hole shapes was also observed in another study with black-tufted marmosets [7]. It has been suggested that elongated scarification is associated with softer bark having superficial secretory structures, where it is more advantageous for marmosets to increase the surface area of the wound [7]. Faria [7] hypothesized that round shapes would be made in trees with harder bark and deeper ducts. In the present study, most of the holes were found in the canopy, where the bark is softer and the ducts are more easily reached. Therefore, it seems that the preference of the marmosets in the present study to make round holes was not related to the ease of reaching deeper layers in harder tree bark. We raised an alternative hypothesis, which we will explain next.

The exudate from *T. guianensis* appears to be of a mixture of gums and resin, as was recently proposed for the Anacardiaceae family [28]. The Hagen-Poiseuille Equation was inadequate for understanding the flow of exudates in bark wounds, as gum-resins do not behave like Newtonian liquids [29]. Therefore, Cabrita [29] constructed a new equation that takes into account the pressure in the duct lumen, the approximate cylindrical geometry of resin duct systems in plants, and the granulocrine loading of resin. According to this model, the larger the exposed surface, the greater the exposure to air and the lower the pressure in the ducts that determine the flow.

After being secreted by the tree, the exudate comes into contact with the air and polymerizes, hardening [28]. Marmosets consume only liquid exudates, rejecting those with a hard consistency [12]. According to hydrodynamic principles, a large-diameter orifice, as seen in elongated wounds, allows for greater contact of the exudate with the air, in addition to producing lower pressure within the ducts [29]. Elongated holes seem to be of little advantage for foraging, as there would be lower volume and faster desiccation of exudate due to greater exposure to the air, resulting in low flow and fast polymerization. A rounder and deeper orifice reduces exudate exposure to air, preventing polymerization. With a smaller exposure area, pressure in the ducts is maintained, allowing the exudate to remain viscous for a longer time, which is suitable for ingestion by the marmoset. Although the shapes of the holes were not recorded according to tree species, it is likely that marmosets scarify more rounded shapes in order to maintain the viscosity that is best adapted to their ingestion capacity. However, this hypothesis needs to be tested in future studies.

The use of non-native species from the Cerrado, such as Brazilian mahogany (*Swietenia macrophylla*), a tree native to the Amazon, and the almond tree (*Terminalia catappa*), an exotic species originating in Asia, corroborates studies by other authors who claim that marmosets can exploit non-native species to obtain exudates [13]. However, given the importance of exudate as a food source for black-tufted marmosets, species such as *T. guianensis* and *C. urucarana* could be used when managing fragments of degraded areas where marmosets occur, helping to reconstitute the reduced native vegetation and guarantee key food resources for marmosets.

Marmosets can exploit many species with a few holes but concentrate the production of exudate from only a few tree species [13]. Ten tree species had only one individual that had been used by marmosets. Other authors also observed a singular use for nine tree species by black-tufted marmosets [11]. It is unclear why marmosets use only one or two individuals of some tree species, which have few holes, and often do not exude enough. Some authors have suggested that these occasional scarifications might be an exploratory activity to probe the exudate resources in the territory [4]. Lazaro-Perea and collaborators [30] found that common marmosets (*C. jacchus*) mark with urine and rub their anogenital glands shortly after gouging, to communicate with individuals both within and outside the group. Scent-marking behavior is common in free-ranging and captive marmosets and serves as communication between individuals [5,30]. Thus, we suggest that the occasional scarification of trees that do not exudate has a double function: to leave marks for intraspecific communication and to execute resource (exudate) prospection.

Marmosets preferred to exploit trees with intermediate DBH and height, which seem to be advantageous to the animal-plant relationship. Larger and older trees, despite the high degree of use, showed predominantly older injuries, suggesting that marmosets may abandon exploration after a tree reaches a certain age. As the trees mature, the marmosets seem to abandon them, allowing for healing of injuries and, most likely, greater tree longevity.

A population of hybrid marmosets, which mainly exploit *A. peregrina* and utilized exudate trees at different intensities, had already been observed [16]. These authors classified the trees using a DBH system of 10 cm measures (10, 20, 30 cm etc.), noting that exploitation was the most intense for trees in the 20, 40, and 60 cm DBH categories, alternating with less-used 10, 30, and 50 cm DBH trees [16]. DBH is related to the age of the tree [31], which led the authors to suggest that there is a cycle of use according to the maturity of *A. peregrina* trees, which can live for 30 years or more [32]. As wild marmosets live for around 5–7 years [33], the trees thus exploited remained available for use by several generations of marmosets [16]. It appears that marmosets are sensitive to the condition of trees, exploiting them in a manner that maintains the availability of food sources.

Some authors [34] observed that the number of dead trees with marmoset scars was greater than dead trees without scars in a forest. Those authors proposed that marmosets increase the turnover of trees exploited for exudate by marmosets in small forests. This evidence, together with their role as seed dispersers [35], suggests that marmosets are true ecosystem “engineers”, as has been proposed for other mammal species (e.g., *Canis lupus* [36]; *Castor canadensis* [37]).

This study made it possible to propose a new approach to evaluate the effort required to exploit exudate trees. ChiSI can be adopted as a proxy for determining the effort of marmosets in tree-gouging, with values determined for the most commonly used species (*T. guianensis*, and *C. urucurana*) being very close to one another. The other species studied had ChiSI values either above or below the limits for the preferred species, *T. guianensis* and *C. urucurana*. This suggests that the trees most often explored for their exudate must have bark that fits within essential density and thickness limits that suit the anatomical and functional limits of the oral apparatus of marmosets. These limits do not appear to be merely physical, because trees with a lower ChiSI were also less frequently used. Trees with a higher ChiSI required more effort to gouge, resulting in a higher energy cost, though they eventually produced more exudates than trees with a lower ChiSI. Additionally, a tree with “sub-optimal” ChiSI can increase the time the individual will remain gouging, increasing its vulnerability to predators. On the other hand, we do not rule out the possibility that the exploitation of trees was linked to other factors. For example, it is well known that the exploitation of exudates is linked to the nutritional needs of marmosets, such as calcium and complex carbohydrates [5,15]. Selection for trees that have nutrient-rich exudates for the black-tufted marmoset’s diet is an approach that could be investigated in the future.

## 5. Conclusions

Several exudate tree species were used by black-tufted marmosets in the Cerrado, even in urbanized forest fragments. Eleven more tree species used by black-tufted marmosets for exudates were identified in urban forests in the Cerrado. The exploitation of exudates is of medium intensity, concentrated in the canopy, and obtained through round holes. The density and thickness of the bark seem to be limiting factors for the exploitation of exudates. In this case, *T. guianensis* and *C. urucarana* were the two species most often exploited by black-tufted marmosets. These two tree species should be considered the most important for the management and preservation of trees for the black-tufted marmoset populations.

In addition to identifying the species, our study raises some questions about the use of exudates trees by marmosets. We proposed a gouge-effort index (ChiSI) to assess marmosets’ preference for some tree species. We suggested an alternative hypothesis based on Fluid Physics for the shapes of holes caused by marmosets in the bark of trees. We also confirmed that marmosets, along with a new undescribed tree species, the *C. urucarana*, prefer *T. guianensis*. We also proposed new descriptions of tree species exploited by marmosets to obtain exudates. Therefore, our study has heuristic value for further research into the relationship between marmosets, the exudativory, and trees.

## Figures and Tables

**Figure 1 animals-12-02578-f001:**
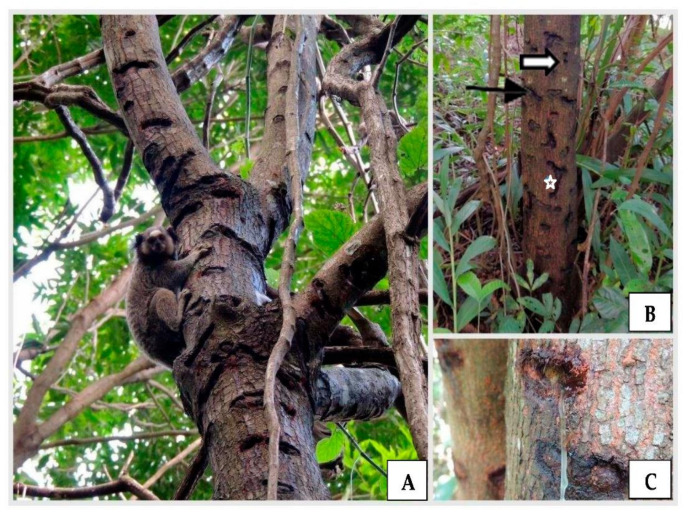
(**A**) Use of *Tapirira guianensis* tree as a source of exudate for a black-tufted marmoset (*Callithrix penicillata*). (**B**) Several orifices, with different shapes—round (white arrow), elongated (black arrow) and irregular (star)—distributed throughout the trunk. (**C**) *T. guianensis* exudation.

**Figure 2 animals-12-02578-f002:**
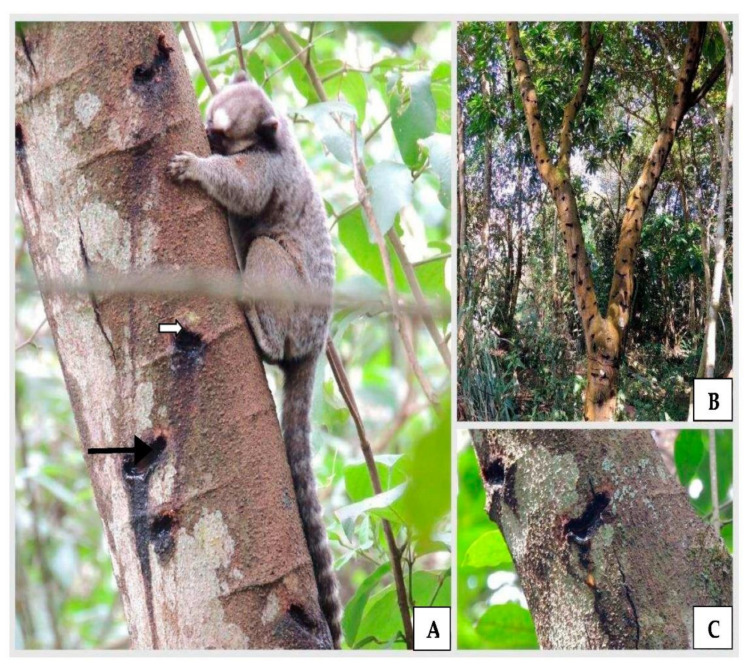
(**A**) The use of the *Croton urucurana* tree as a source of exudate by the black-tufted marmoset (*Callithrix penicillata*). Round (white arrow) and elongated (black arrow) shaped holes in the trunk. (**B**) Several holes were distributed throughout the trunk and branches. (**C**) *C. urucurana* exudation.

**Table 1 animals-12-02578-t001:** Tree species used by black-tufted marmosets (*Callithrix penicillata*) to obtain exudates in urban forest fragments in municipality of Goiânia, GO, Brazil. In the columns are the species, botanical families, the relative frequency, exploration characteristics, and known species, with the source in the literature. The list is arranged in descending order of relative frequency. Two unidentified species were removed from the sample. Total of individuals = 108. Trees from Anacardiaceae and Euphorbiacea families had the highest relative abundance (45.4% and 31.5%, respectively).

Species and Family	Relative Frequency (%) and Number of Individuals (n)	Predominance of Location and Shape of Holes	Exploration Intensity Class	Tree Use and References
*Tapirira guianensis* Aubl. (Anacardiaceae)	44.5 (n = 48)	Trunk and canopy. Round, elongated and irregular	Intense	Faria (1983); Santee and Faria (1985); Lacher et al. (1984); Vilela (2007); Vilela and Del Claro (2011)
*Croton urucurana* Baill. (Euphorbiaceae)	31.5 (n = 34)	Trunk and canopy. Round and elongated	Moderate	Species not described in scientific articles
*Anadenanthera* sp. (Fabaceae)	8.4 (n = 9)	Canopy. Round	Intense	Miranda and Faria (2001)
*Terminalia glabrescens* Mart. (Combretaceae)	3.7 (n = 4)	Canopy. Round	Moderate	Species not described in scientific articles
*Handroanthus* *impetiginosus* (Mart. ex DC.) Stand. (Bignoniaceae)	2.0 (n = 2)	Canopy. Round	Light	Species not described in scientific articles
*Anacardium occidentale* L. (Anacardiaceae)	0.9 (n = 1)	Canopy. Round	Light	Species not described in scientific articles
*Xylopia emarginata* Mart. (Annonaceae)	0.9 (n = 1)	Trunk. Round	Light	Species not described in scientific articles
*Enterolobium* *contortisiliquum* (Vell.) Morong. (Fabaceae)	0.9 (n = 1)	Canopy an trunk. Round	Intense	Vilela and Faria (2002)
*Mimosa* *caesalpiniaefolia* Benth. (Fabaceae)	0.9 (n = 1)	Trunk. Round	Moderate	Species not described in scientific articles
*Piptadenia gonoacantha* (Mart.) Macbr. (Fabaceae)	0.9 (n = 1)	Canopy. Round	Moderate	Vilela and Faria (2002)Zago et al. (2013)
*Guarea guidonia* (L.) Sleumer (Meliaceae)	0.9 (n = 1)	Canopy. Round and irregular	Light	Species not described in scientific articles
*Swietenia macrophylla* King. (Meliaceae*)	0.9 (n = 1)	Canopy Round	Intense	Species not described in scientific articles
*Triplaris gardneriana* Wedd. (Polygonaceae)	0.9 (n = 1)	Canopy. Round	Moderate	Species not described in scientific articles
*Zanthoxylum rhoifolium* Lam. (Rutaceae)	0.9 (n = 1)	Trunk. Round and elongated	Moderate	Species not described in scientific articles
*Qualea dichotoma* (Warm.) Stafl. (Vochysiaceae)	0.9 (n = 1)	Canopy. Round	Moderate	Lacher et al. (1984) Vilela (2007)
*Styrax camporum* Pohl. (Styracaceae)	0.9 (n = 1)	Trunk. Elongated and irregular	Moderate	Species not described in scientific articles

**Table 2 animals-12-02578-t002:** Bark density, thickness and ChiSI (Chiseling Suitability Index) of exudating trees, exploited by black-tufted marmosets (*Callithrix penicillata*), in three urban forest fragments of the municipality of Goiânia, GO, Brazil. Density and thickness are expressed in mean (±standard deviation). The ChiSI is a product between the density × thickness, and there is no derivative metric unit to express it.

Species	Density (g/cm^3^)	Thickness (mm)	ChiSI
*Anadenanthera* sp.	0.76 ± 0.08	11.51 ± 3.74	8.75
*Qualea dichotoma*	0.58 ± 0.02	9.80 ± 0.68	5.70
*Guarea guidonia*	0.42 ± 0.01	9.86 ± 1.12	4.14
*Handroanthus impertiginosus*	0.43 ± 0.02	8.99 ± 0.69	3.68
*Tapirira guianensis*	0.66 ± 0.04	5.07 ± 0.54	3.35
*Croton urucurana*	0.50 ± 0.02	4.89 ± 0.02	2.44
*Triplaris gardneriana*	0.47 ± 0.05	4.66 ± 0.24	2.19
*Zanthoxylum rhoifolium*	0.47 ± 0.07	3.60 ± 0.71	1.70
*Xylopia emarginata*	0.40 ± 0.01	4.13 ± 0.80	1.65
*Anacardium occidentale*	0.34 ± 0.06	3.62 ± 0.47	1.23
*Mimosa caesalpiniifolia*	0.39 ± 0.02	3.11 ± 0.80	1.21

## Data Availability

The data that support the findings of this study are available from the corresponding author, V.B., upon reasonable request.

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
