# Peer review of "Tree Species and Morphology of Holes Caused by Black-Tufted Marmosets to Obtain Exudates: Some Implications for the Exudativory"

_animals, 2022, doi:10.3390/ani12192578_

Round 1
Reviewer 1 Report
Dear Authors
The article submitted for review is very interesting. To its credit is not only the topic and scope of the study, but also the detailed photographic documentation. I would have included in the Materials and Methods section a photo of the skull of a representative of Callithrix penicillata with a close-up especially of the teeth. The authors already mention in the Simple Summary section the important role of the teeth themselves...Black-tufted marmosets (Callithrix penicillata) regularly exploit exudates by gouging the shoulder from trees with their specialized teeth...I would therefore ask the authors to include, in addition to a photo of these specialized teeth, also a brief information on what exactly is the specialization of these teeth? Please also improve the quality of Fig. 1- it is illegible.
Regards
Author Response
Dear reviewer
We thank your comments on the manuscript.
In particular, all changes were marked in color in the tracked manuscript.
We respond to each reviewer's suggestion, marking our comments below.
Please, note that some modifications were made considering the comments of another reviewer as well.

Reviewer 2 Report
This is an interesting and unusual study that is worthy of publication. The study is well planned and executed, the data appropriate, and the discussion follows well with the results. However, the conclusion that “Based on these findings, it appears that the species T. guianensis and C urucana strike a balance between the effort to gouge and the volume of exudates the marmosets can obtain” is somehow not appropriate because you did not evaluate the quantity or quality of the exudates. Reword to conclude what you have really found in your results.
English needs to be edited by a professional and the many grammatical glitches corrected. I give some examples where I started editing but do not have the time to do the whole manuscript. Sorry.
In the discussion, the authors have not accounted for the differences in the nutritional value of the exudates of the different tree species. I understand that this was not evaluated but has it been done by others? Or could this in any manner influence the configuration of the holes that the marmosets make? I think that there is place to include a paragraph about the possibility that there are other, hitherto unevaluated, parameters that could influence the hole gouging behavior.
In lines 322-326 you mention the marking with urine and anogenital glands for intra-specific communication. I think this point needs to be expounded upon and examples of this behavior in other species given to illustrate the wide use of this technique across groups.
I am missing a take-home message with a conservation recommendation as to how and why the tree species you have identified should be given special consideration/protection.
Simple Abstract:
“The values of the ChiSI of these two preferred species are close, with no ChiSi values of the other trees species between them.” – Delete
“The black-tufted marmosets appear to concentrate the exudate exploration to a few tree species.” Change
Abstract:
“Of the 108 individualstrees explored by black-tufted marmosets, 16 species belonginged to 10 tree families were found.”
“Eleven new 42 tree species used by bBlack-tufted marmosets …….” – Please be consisten throughout the manuscript.
“ …… marmosets concentrates exploitation to a limited variety of tree species.”
Introduction:
Line 75: “density (i.e., the ratio of……”
Line 79: “The bark of the exploited parts of the tree should is not be an impeditive barrier …..”
Line 80: : …. morphological and physiological capacity capabilities of the marmosets' ….”
Line 111 – What is Aw? Spell out before you give the acronym.
Fig.1. I hope you have a high-resolution picture for the final product. It is pretty fuzzy.
Line 160: “ …… a 150mm Digital 6 Pol. Western digital caliper.” + how much?
Author Response

(The authors gave the same response as above.)

Round 2
Reviewer 2 Report
The revised version is much improved and a pleasure to read. However, there are a few glitches that I suspect will be ironed out in the editorial process. A few suggestions to change wording:
Title: TREE SPECIES AND MORPHOLOGY OF HOLES CAUSED BY BLACK-TUFTED MARMOSETS TO OBTAIN EXUDATES: SOME IMPLICATIONS FOR THE EXUDATIVORY
Line 61: Marmosets of the species C. jaccchus have a….. (period is missing after C)
Line 62: … and have a relatively wide jaw opening .. (since you are not making a comparison it is not “relative to ..“)
Lines 257-259: should not be in the text J - “This section may be divided by subheadings. It should provide a concise and precise description of the experimental results, their interpretation, as well as the experimental conclusions that can be drawn.”
Line 275: Our results are in line concur with……..
Line 402: These other factors may should be investigated in the future.
Author Response
Reply to reviewer regarding the manuscript “Tree Species and Hole Morphology Caused by Black-Tufted Marmosets to Obtain Exudates: Some Implications for the Exudativory”
Dear reviewer
Thank you for taking the time to review our manuscript. We thank your comments on the manuscript. In particular, all changes were marked in color in the tracked manuscript. We respond to each reviewer's suggestion, marking our comments below.
Line 61: Marmosets of the species C. jaccchus have a….. (period is missing after C)
Response: We modified the sentence – “Marmosets of the C. jaccchus species have…”
Line 62: … and have a relatively wide jaw opening .. (since you are not making a comparison it is not “relative to ..“)
Response: We modified the sentence – “Marmosets of the C. jaccchus species have a bite force of up to eight times their body mass and have a wide jaw opening”.
Lines 257-259: should not be in the text J - “This section may be divided by subheadings. It should provide a concise and precise description of the experimental results, their interpretation, as well as the experimental conclusions that can be drawn.”
Response: Surely this sentence should not be in the text. We apologize. In the process of copying and pasting, we somehow made this bizarre mistake, which we deleted.
Line 275: Our results are in line concur with……..
Response: We modified the sentence – “Our results are in agreement with other studies …”
Line 402: These other factors may should be investigated in the future.
Response: We modified the sentence – “Selection for trees that have nutrient-rich exudates for black-tufted marmoset diet is an approach that could be investigated in the future”.
